# Computationally efficient design of directionally compliant metamaterials

Lucas A. Shaw[1], Frederick Sun[1], Carlos M. Portela[2], Rodolfo I. Barranco[1], Julia R. Greer[2] & Jonathan B. Hopkins [1]

Designing mechanical metamaterials is overwhelming for most computational approaches because of the staggering number and complexity of flexible elements that constitute their architecture—particularly if these elements don't repeat in periodic patterns or collectively occupy irregular bulk shapes. We introduce an approach, inspired by the freedom and constraint topologies (FACT) methodology, that leverages simplified assumptions to enable the design of such materials with ~6 orders of magnitude greater computational efficiency than other approaches (e.g., topology optimization). Metamaterials designed using this approach are called directionally compliant metamaterials (DCMs) because they manifest prescribed compliant directions while possessing high stiffness in all other directions. Since their compliant directions are governed by both macroscale shape and microscale architecture, DCMs can be engineered with the necessary design freedom to facilitate arbitrary form and unprecedented anisotropy. Thus, DCMs show promise as irregularly shaped flexure bearings, compliant prosthetics, morphing structures, and soft robots.

[1] Mechanical and Aerospace Engineering, University of California, Los Angeles, Los Angeles, CA 90095, USA. [2] Division of Engineering and Applied Science, California Institute of Technology, Pasadena, CA 91125, USA. Correspondence and requests for materials should be addressed to J.B.H. (email: hopkins@seas.ucla.edu)

Mechanical metamaterials (a.k.a. architected materials) can achieve extreme properties that derive primarily from their architecture instead of their composition[1]. By controlling the locations and orientations of microelements (e.g., beams, blades, and hinges) that constitute their architecture, such materials can be engineered with super properties otherwise not achievable (e.g., extreme strength-to-weight ratios[2], tunable negative thermal expansion coefficients[3], and large negative Poisson's ratios[4]).

Past research has primarily focused on infinite periodic metamaterials that achieve their engineered properties with isotropy because such materials consist of single symmetric cells that repeat without bounds and are thus manageable to design despite their numerous constituent elements. Unfortunately, such metamaterials have limited use because most practical applications require materials that occupy finite and often irregularly shaped volumes and achieve anisotropic properties tailored along prescribed directions. Metamaterials that meet these demands usually require huge numbers of aperiodic (i.e., nonrepeating) asymmetric cells that occupy volumes with complex boundaries and are thus too computationally expensive to design.

Previous work has sought to address these challenges by utilizing precomputed databases of different cell designs to generate aperiodic and practically shaped metamaterials that achieve desired deformations[5] or targeted regions of compliance[6]. Finite element analysis (FEA), sparse regularization, and constraint optimization have been employed to generate shapes consisting of aperiodic distributions of different materials that deform in prescribed ways when actuated[7]. Aperiodic metamaterials have also been designed with graded properties (e.g., elasticity[8] and thermal expansion[9]), which vary across their lattice's geometry. Additionally, metamaterials that exhibit desired textures when actuated have been designed using a single anisotropic cell that is oriented in nonrepeating patterns[10]. Lastly, aperiodic lattices of shearing cells have been used to generate monolithic mechanisms that achieve desired deformations[11].

Despite these advances, a large computational gap remains between metamaterial research and the ability to implement that research within most practical applications. A new approach is necessary to bridge this gap by leveraging simplified assumptions to enable the automated design of aperiodic metamaterials of staggering complexity and achieve customized anisotropic properties while assuming any form.

Metamaterials designed using this approach are called directionally compliant metamaterials (DCMs) because they are engineered to achieve high compliance along desired directions while exhibiting high stiffness along other directions. In contrast with traditional flexure systems[12], which are currently used to achieve desired directions of compliance (i.e., degrees of freedom (DOFs)), DCMs can be engineered to assume any bulk shape while achieving unprecedented combinations of DOFs. The reason is that unlike flexure systems, which achieve DOFs almost exclusively according to how they are shaped on the macroscale, DCMs achieve their anisotropic properties both according to their macroscale shape as well as their architecture at smaller scales. Thus, the design space of DCMs that achieve desired DOFs while simultaneously assuming desired bulk shapes is significantly larger than the design space of flexure systems that achieve the same objectives.

An example that demonstrates these advantages is a prosthetic elbow joint. Although flexure systems (e.g., Fig. 1a) could achieve the joint's desired rotational DOF with high compliance while possessing high stiffness in all other directions, no flexure system could also assume the irregular shape of an elbow. An aperiodic DCM (e.g., Fig. 1b) could, however, achieve the desired rotational DOF (Fig. 1c) while also assuming an elbow shape. Such a joint would avoid the need for assembly and could be additively fabricated as a monolithic structure while mimicking an elbow with greater practicality and fidelity.

In addition to enabling directionally compliant joints, DCMs can facilitate other shape-morphing applications. A DCM could, for example, be shaped on the macroscale as a propeller blade but be engineered with a microarchitecture that exhibits a screw DOF (i.e., a translation coupled with a rotation)[13] about the blade's axis while achieving high stiffness in all other directions. The pitch of the screw DOF could be tuned such that its corresponding blade would passively reconfigure its angle of attack proportionate to the angular speed of the propeller due to centripetal forces. Other DCM applications are discussed in Supplementary Note 1 and shown in Supplementary Fig. 1 and Supplementary Movie 1.

Most DCMs are currently impossible to design because their architecture typically consists of unmanageably large numbers of nonrepeating flexible elements that collectively occupy irregularly shaped volumes. Existing computational approaches (e.g., topology optimization[14]) become overwhelmed when searching the design space of DCMs because the space is infinitely large and the process of searching the space requires the simultaneous optimization of huge numbers of parameters.

This paper introduces the theory necessary to design arbitrarily shaped DCMs that are locally comprised of easily computed anisotropic constituents. Inspired by the mathematics underlying the freedom and constraint topologies (FACT) approach[15–17], this theory leverages simplified assumptions about constituent elements to enable the automated design of three-dimensional (3D) DCMs of immense complexity with unmatched efficiency. We demonstrate the theory's computational superiority using our MATLAB tool (see Supplementary Software) and introduce the principles that govern how both macroscale form and architecture affect the DOFs of DCMs.

## Results

**Design approach**. The approach introduced here leverages the vector spaces of the FACT library[15–17] graphically depicted in Supplementary Fig. 2 to rapidly generate DCMs with desired DOFs. The vector spaces of the FACT library utilize screw theory[18–20] and collectively embody the design space of all compliant systems. One set of spaces, called freedom spaces[15–17], consist of red rotation lines, green screw lines, and black translation arrows and represent all the combinations of DOFs that a system could achieve. Another set of complementary spaces, called constraint

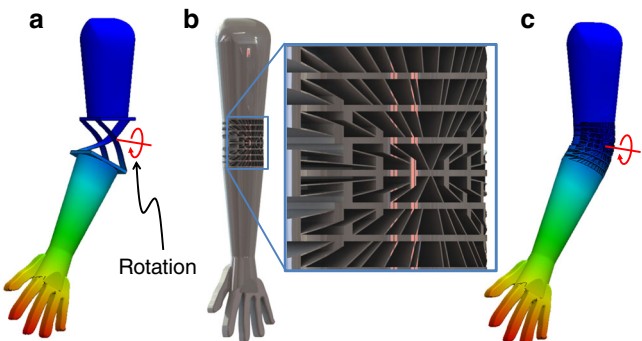

**Fig. 1** Introduction to directionally compliant metamaterials (DCMs). **a** A flexure system that exclusively achieves the desired rotational degree of freedom (DOF) of an elbow but fails to assume its shape and, **b** an aperiodic DCM that can be shaped to conform to any elbow shape while also achieving high stiffness in all directions except about, **c** the desired compliant rotational DOF

spaces[15–17], consist of blue constraint-force lines and represent the region of space within which flexible elements must be placed to achieve the DOFs of their corresponding freedom space. Additional FACT-library details are discussed in Methods.

Although the FACT library was originally created to facilitate the synthesis of flexure systems via a paper–pencil approach, this work demonstrates that an advanced automated approach can leverage the same library alongside computation to enable the rapid generation of complex DCMs with even greater benefit to the field of metamaterials. Whereas other approaches fail to generate DCMs because their computational cost is too high, the approach introduced here can generate DCMs with orders of magnitude less cost. The reason is that unlike other computational approaches that simultaneously consider the constituent material properties, geometric parameters, locations, and orientations of every element within a DCM, the approach introduced here simplifies the scenario significantly by only modeling the locations and orientations of each element using $6 \times 1$ pure-force wrench vectors (PFWVs)[15–20], $\mathbf{W}_{6 \times 1}$. These vectors are depicted as the blue constraint-force lines within the constraint spaces of the FACT library in Supplementary Fig. 2. The mathematics required to define PFWVs and to use these vectors to model elements of any geometry are provided in Methods.

Elements modeled using PFWVs are treated as ideal elements that are infinitely stiff along the axes of the blue constraint-force lines that pass through the element's geometry but are infinitely compliant in all other directions. This assumption dramatically simplifies the design process such that the locations and orientations of hundreds to thousands of elements per second can be determined within DCMs using a standard desktop computer. Although the ideal-element model produces DCMs that would theoretically exhibit infinite stiffness in all directions except along their infinitely compliant DOFs, once geometric parameters and material properties are assigned to their elements, the DOFs achieved by such DCMs actually exhibit finite compliant values that are consistently the most compliant of all other directions.

The proposed approach's steps are briefly summarized here. The DCM's volume is first divided into smaller cell volumes within which elements are to be placed to ensure that each cell will individually achieve the desired DOFs. A DCM consisting of many such smaller cells could be made to assume a variety of bulk shapes without compromising the desired DOFs because each cell is redundant and can, therefore, be removed from the material's volume with minimal consequence. Once the DCM's volume is divided into constituent cell volumes, the DCM's desired DOFs are then modeled as $6 \times 1$ twist vectors[15–20], $\mathbf{T}_{6 \times 1}$, according to the mathematics detailed in Methods. The freedom space that represents the combination of all the desired DOFs is then calculated by linearly combining the twist vectors that model each DOF. The complementary constraint space of the resulting freedom space is then identified using the FACT library. If this constraint space belongs within the region shaded yellow in the FACT library of Supplementary Fig. 2 (i.e., 0 DOF Type 1, 1 DOF Type 1 through 3, 2 DOF Type 3 through 9, and 3 DOF Type 2 and 3), the geometry of that constraint space can be used to determine the appropriate kind, number, location, and orientation of flexible elements within each cell volume according to the theory in Methods. Such constraint spaces that lie within the yellow shaded region of the FACT library are called cell spaces because they are the only constraint spaces that can occupy any volume of space with enough independent PFWVs to generate cell topologies that achieve their intended DOFs. Thus, if the desired freedom space's complementary constraint space is not a cell space, it can't be used to synthesize the DCM's cells. As a result, alternating layers of cells that each achieve some of the DOFs within the freedom space should be designed to collectively achieve all the DOFs within the freedom space when they are stacked together in series. To synthesize such serially-stacked layers, intermediate freedom spaces[16,17] should be selected from within the freedom space according to the rules in Methods. Each intermediate freedom space selected represents the combination of the DOFs that each serially-stacked cell layer will contribute to the DCM's freedom space. The intermediate freedom spaces selected must link to complementary constraint spaces that are cell spaces because these spaces must then be used to generate the individual cell topologies within the DCM's alternating cell layers.

A case study of the design approach is provided here and animated in Supplementary Movie 2. The case study is a DCM that achieves a single screw DOF with a desired pitch, $p$, as shown by the green line in Fig. 2a. The DCM volume is first divided into individual cell volumes as shown. The freedom space of the desired screw DOF is then identified as the freedom space labeled 1 DOF Type 2 in Supplementary Fig. 2 (Fig. 2b). Its constraint space consists of nested circular hyperboloids filled with PFWVs that satisfy $p = d \cdot \tan(\theta)$ according to the geometric parameters, $d$ and $\theta$, labeled in Fig. 2b. Since the constraint space is a cell space (i.e., it belongs within the region shaded yellow in Supplementary Fig. 2), each cell that constitutes the final DCM design (Fig. 2c) is synthesized from within the geometry of the constraint space according to the rules provided in Methods. The resulting DCM consists of nine identical stacked layers (Fig. 2d) constructed using six different cell designs (Fig. 2e) that each utilize five wire elements (i.e., slender cylindrical beams) aligned with independent PFWVs from within the constraint space of Fig. 2b. Although this space contains enough independent PFWVs that pass through the volume of each cell within the DCM because the space is a cell space, not all of the PFWVs' corresponding colinear wire elements can directly join the cell's rigid bodies together without layer extensions that protrude from these bodies. Thus, layer extensions are used within some of the cell designs (i.e., the blue, green, yellow, and red cells in Fig. 2e). If a higher cell resolution had been specified such that many more cells would have been generated, a propeller-blade shape could have been carved out of the resulting DCM without altering its screw DOF to enable the propeller application discussed previously.

We fabricated the DCM of Fig. 2c at the microscale using two-photon lithography, which achieved minimum feature resolutions of ~1.5 μm (Fig. 3a). To validate the desired screw DOF, we performed in situ uniaxial compression experiments (Supplementary Movie 2) while tracking the rotation of each rigid layer using a scanning electron microscope (SEM). Imposing quasi-static deformation ($\dot{\varepsilon} = 10^{-3}$ s$^{-1}$) to the elastic strain limit ($\varepsilon \approx 8\%$) produced the corresponding clockwise rotation according to the intended pitch of the desired screw DOF. This elastic response was validated via FEA (Fig. 3b), which showed the same rotation upon compression. The details of this FEA are specified in Methods. The FACT-predicted pitch, $p$, of 30 μm/rad was closely matched by the FEA calculations, while the experiments achieved an average pitch of 38.3 μm/rad, attributed to non-negligible friction between the indenter and the top pyramid-shaped layer as well as inherent manufacturing defects (Fig. 3c). To assess the repeatability of the screw deformation, we performed cyclic compressions (Fig. 3d) in which a constant pitch was observed above a ~4 μm displacement. Minor permanent deformation accumulated after the first two cycles, which prevented the material to revert to the zero-rotation state upon unloading, but it did not affect the value of the pitch when deformed in the linear regime. Additional plots are provided in Supplementary Fig. 3. Note that although an alternative single-screw-DOF metamaterial has previously been designed[13] prior to this work, the theory of this paper enables the automated

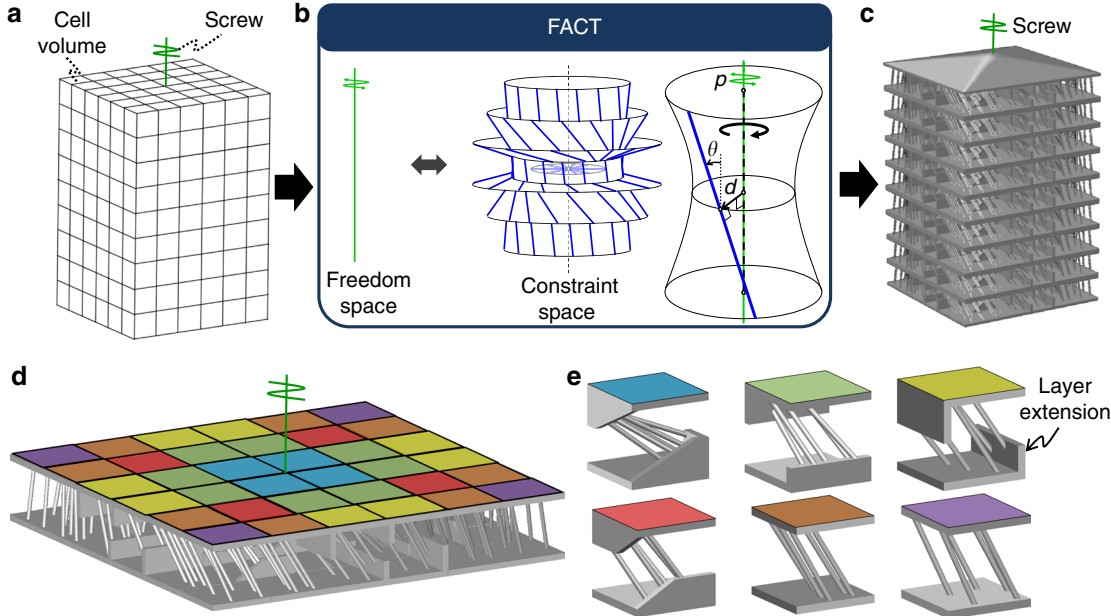

**Fig. 2** Single degree-of-freedom (DOF) screw example. **a** The available volume is divided into individual cell volumes and the desired screw DOF is specified. **b** The screw DOF's freedom space and its complementary constraint space shown with parameters that relate its geometry to the pitch, *p*, of the screw, **c** the resulting aperiodic directionally compliant metamaterial (DCM) design consisting of, **d** nine identical layers each made of, **e** six unique cell designs

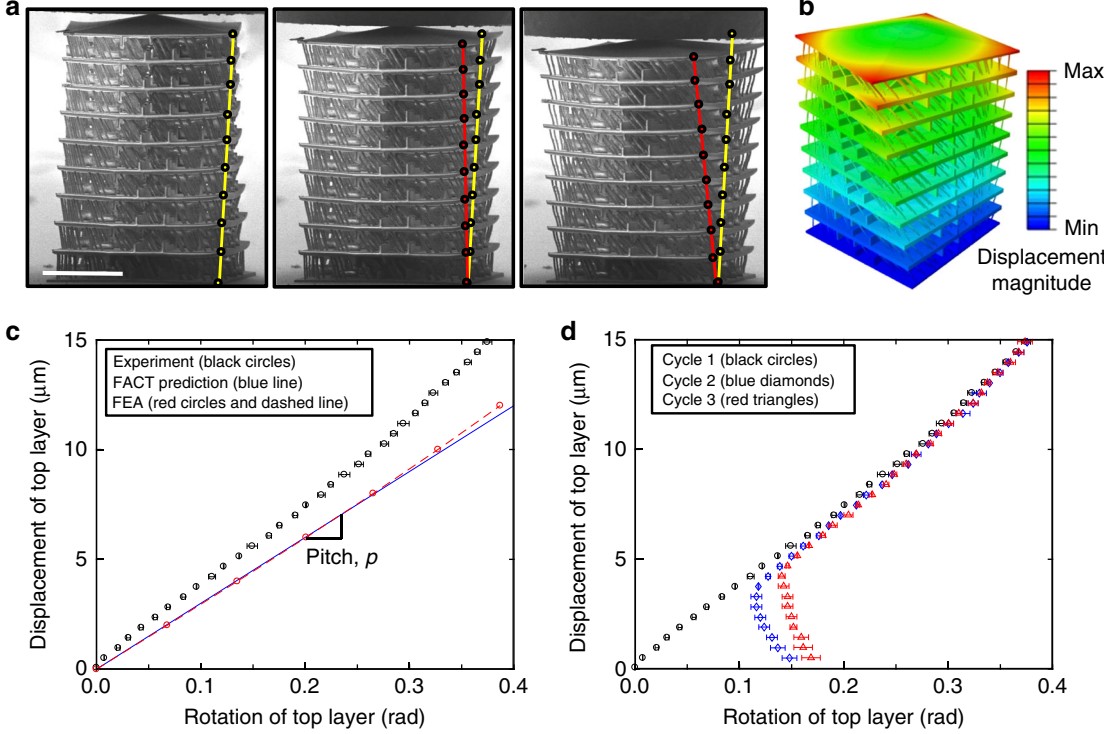

**Fig. 3** Validation of the screw degree-of-freedom (DOF) example. **a** In situ nanomechanical compression experiment on a screw directionally compliant metamaterial (DCM) fabricated using two-photon lithography, during which the corners of the rigid layers were tracked (red circles) and compared with the undeformed configuration (yellow circles). **b** Finite element analysis (FEA) assuming fully linear behavior depicting the clockwise rotation observed in experiments. **c** Pitch comparison between freedom and constraint topologies (FACT) prediction and FEA (30 μm/rad), and experiments (38.3 μm/rad). **d** Cyclic compression of the screw-DOF DCM. The experimental data points in **c** and **d** correspond to the averaged top-layer rotation for a minimum of two identical samples, while the error bars correspond to the standard deviation amongst the samples (scale bar in **a**, 50 μm)

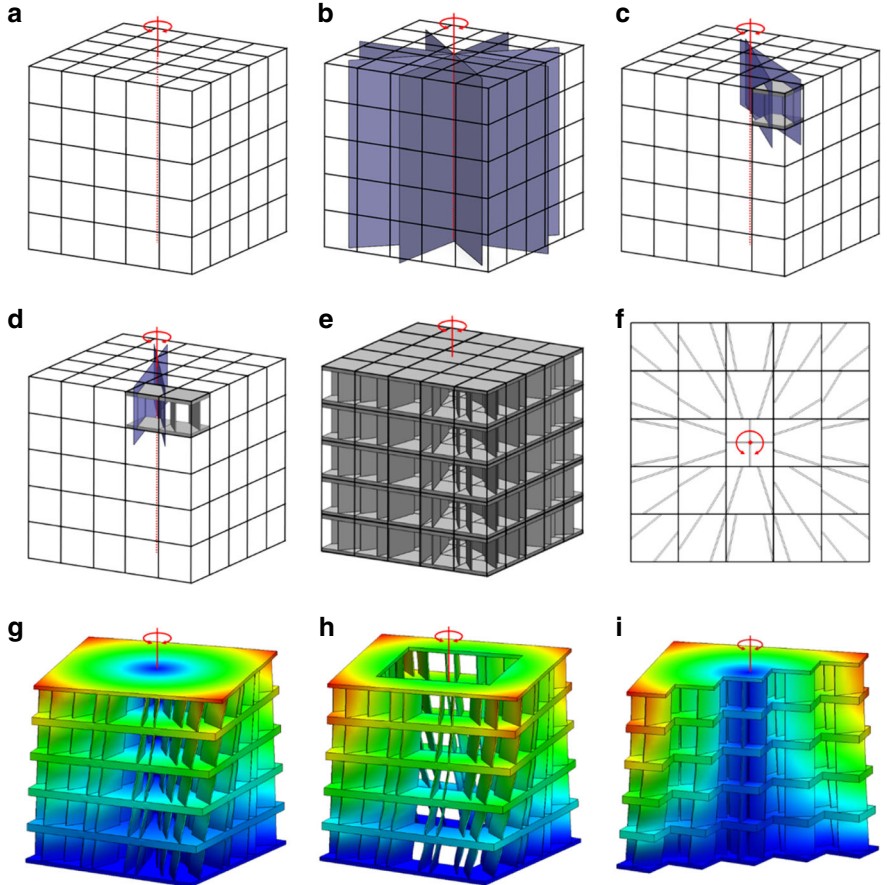

**Fig. 4** Example with a rotational degree-of-freedom (DOF). **a** A rotational freedom space and, **b** its constraint space of intersecting planes can be used, **c**, **d**, to synthesize individual cells that individually achieve the desired rotation so, **e**, an aperiodic directionally compliant metamaterial (DCM) with, **f** intersecting blade elements can be generated to achieve, **g–i**, the desired rotation for a variety of bulk shapes (colors in **g–i** are defined in Fig. 3b)

synthesis of metamaterials that achieve any combination of DOFs (i.e., screws, translations, and rotations) located and oriented any way desired.

**Single-DOF case study**. Suppose a cube-shaped 5×5×5-cell DCM is desired that is stiff in all directions except about a single rotational axis through its center as shown in Fig. 4a. The freedom space that embodies the desired rotational DOF is depicted as the red line, labeled 1 DOF Type 1 in Supplementary Fig. 2. Its constraint space consists of the intersecting blue planes shown in Fig. 4b. Since this constraint space is a cell space, the portion of the space that fills each cell volume can be used to synthesize their respective topologies. Two blade elements per cell can, for instance, be selected such that each blade's plane corresponds with a plane from the constraint space as shown in Fig. 4c, d to ensure that each cell individually achieves the desired DOF. Recall that the rules for determining the number and way flexible elements should be selected from within constraint spaces to achieve the desired DOFs embodied by their freedom spaces are provided in Methods. The remaining cells can be similarly synthesized to generate the aperiodic DCM of Fig. 4e. Note from the view shown in Fig. 4f that the planes of the blade elements all intersect the rotational axis. Modal analysis demonstrates that regardless of what constituent material the resulting design is assigned, the first mode shape corresponds with the desired compliant rotation (Fig. 4g) for a variety of DCM bulk shapes, e.g., a hollowed-out cube (Fig. 4h) or a halved cube (Fig. 4i). Many more irregular

shapes (e.g., the elbow shape of Fig. 1b,c) could be carved out of the cube-shaped DCM without compromising its desired rotational DOF if a higher cell resolution is applied. The process for designing this case study is animated in Supplementary Movie 3 and details regarding its FEA verification are provided in Methods.

**Multi-DOF case study**. It is not always obvious which freedom space maps to a given set of DOFs when more than one DOF is desired. Suppose a cube-shaped 4×4×4-cell DCM is desired that achieves the three rotational DOFs shown in Fig. 5a. The freedom space that represents the combination of these intersecting rotations, labeled 3 DOF Type 3 in Supplementary Fig. 2, is the sphere of all red rotation lines that intersect a common point as shown in Fig. 5b. To determine this freedom space, the desired DOFs were modeled using twist vectors according to the theory in Methods and were linearly combined to generate all the other twist vectors within the resulting freedom space. The freedom space's complementary constraint space is a sphere of PFWVs that intersect the same point as the rotation lines within the freedom space. Since this constraint space is a cell space, the DCM of Fig. 5c could be synthesized by aligning the axes of three wire elements in each cell with three independent PFWVs from within the constraint space of Fig. 5b according to the rules detailed in Methods. Note that many of the resulting cell designs require layer extensions. Regardless of constituent material properties, the final DCM's first three mode-shapes correspond with the three

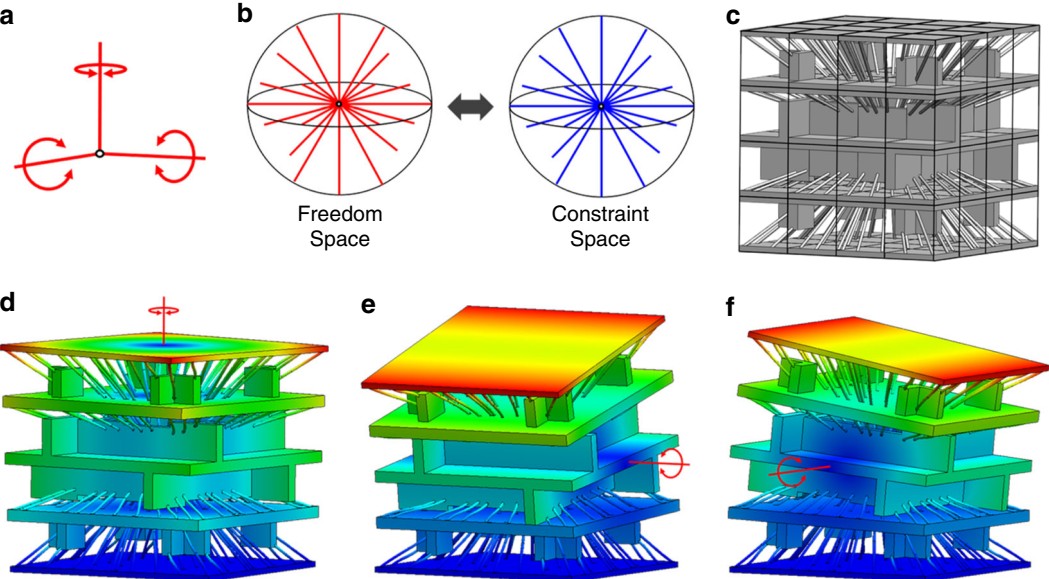

**Fig. 5** Example with three intersecting rotational degrees of freedom (DOFs). **a** Three desired intersecting rotational DOFs and, **b** the freedom and constraint spaces used to synthesize, **c** a directionally compliant metamaterial (DCM) that, **d**–**f**, achieves the desired compliant rotations (colors in **d**–**f** are defined in Fig. 3b)

desired rotations as shown in Fig. 5d–f. If a higher cell resolution had been used, the resulting DCM could have been formed to mimic the DOFs and shapes of natural wrist, shoulder, or hip joints for various prosthetic or soft-robot applications. The process for designing this case study is animated in Supplementary Movie 3 and details regarding its FEA verification are provided in Methods.

**Case study with a freedom space not linked to a cell space**. Not every constraint space can be used to generate the layers of a DCM. Suppose, for instance, a DCM is desired that achieves two intersecting rotations on its top surface as shown in Fig. 6a. The freedom space that represents the combination of those DOFs, labeled 2 DOF Type 1 in Supplementary Fig. 2, is a planar disk of red rotation lines that intersect at the same point (Fig. 6b). Its constraint space consists of a plane of PFWVs that is coplanar with the disk of rotations and a sphere of PFWVs that intersect at the same point where the rotations intersect (Fig. 6c). Since the PFWVs on the plane of the constraint space don't pass through the cells in the DCM, there are not enough independent PFWVs in the rest of the constraint space (i.e., the sphere) to synthesize the cells with flexible elements that directly connect the layers together. Thus, alternating layers of cells that each achieve some of the DOFs within the freedom space should be designed to collectively achieve all the DOFs within the freedom space when they are stacked together. To synthesize such cell layers, intermediate freedom space should be selected from within the freedom space according to the rules discussed in Methods. The intermediate freedom spaces should also link to complementary constraint spaces that are cell spaces since those are the only spaces that can occupy any volume of space with enough independent PFWVs to generate correct cell topologies located anywhere. Note that the freedom spaces of all previous examples link to constraint spaces that are cell spaces but the freedom space of Fig. 6b does not link to a cell space, which is why intermediate freedom spaces that do link to cell spaces are required. Suppose, for this example, the two rotations shown in Fig. 6a were each selected as the intermediate freedom spaces from within the space of Fig. 6b.

The intersecting planes of the first intermediate freedom space's complementary constraint space (Fig. 4b) can be used to synthesize the flexible elements of each cell (Fig. 6d) in the first layer (Fig. 6e) such that the cells in that layer individually and collectively achieve the rotation of their intermediate freedom space. The second intermediate freedom space's complementary constraint space (Fig. 4b) can then be used to synthesize the flexible elements of each cell (Fig. 6f) in the second layer (Fig. 6g) such that the cells in that layer individually and collectively achieve their differently oriented intermediate freedom space. If this process continues for each successive alternating layer, the resulting aperiodic DCM (Fig. 6h) will achieve all the DOFs within the full freedom space of Fig. 6b as shown in Fig. 6i, j. The final design can then be additively fabricated and shaped as desired (Fig. 6k). The process for designing this case study is animated in Supplementary Movie 3 and details regarding its FEA verification are provided in Methods.

**Automated design tool**. A MATLAB tool (provided in Supplementary Software) was created to automate the design of DCMs. The tool first prompts users to specify cell size and resolution. In the example of Fig. 7, a cell size of 2.54 cm and a resolution of 4×4×4 cells was chosen. The tool then prompts users to specify the desired DOFs and to identify their corresponding freedom space. In the example of Fig. 7a, two orthogonal translational DOFs and two orthogonal rotational DOFs were chosen on the top surface of the DCM, which combine to produce the freedom space, labeled 4 DOF Type 8 in Supplementary Fig. 2. This freedom space contains a disk of translations and an infinite number of stacked disks filled with rotations and screws (Fig. 7b). If the freedom space selected links to a constraint space that is a cell space, this constraint space is used by the tool to generate all the cells within the DCM using the mathematics detailed in Methods. If, however, the freedom space does not link to a cell space, the tool then requires the user to identify intermediate freedom spaces that link to constraint spaces that are cell spaces and combine to produce the freedom space. Since the freedom space in Fig. 7b does not link to a cell space, the freedom space

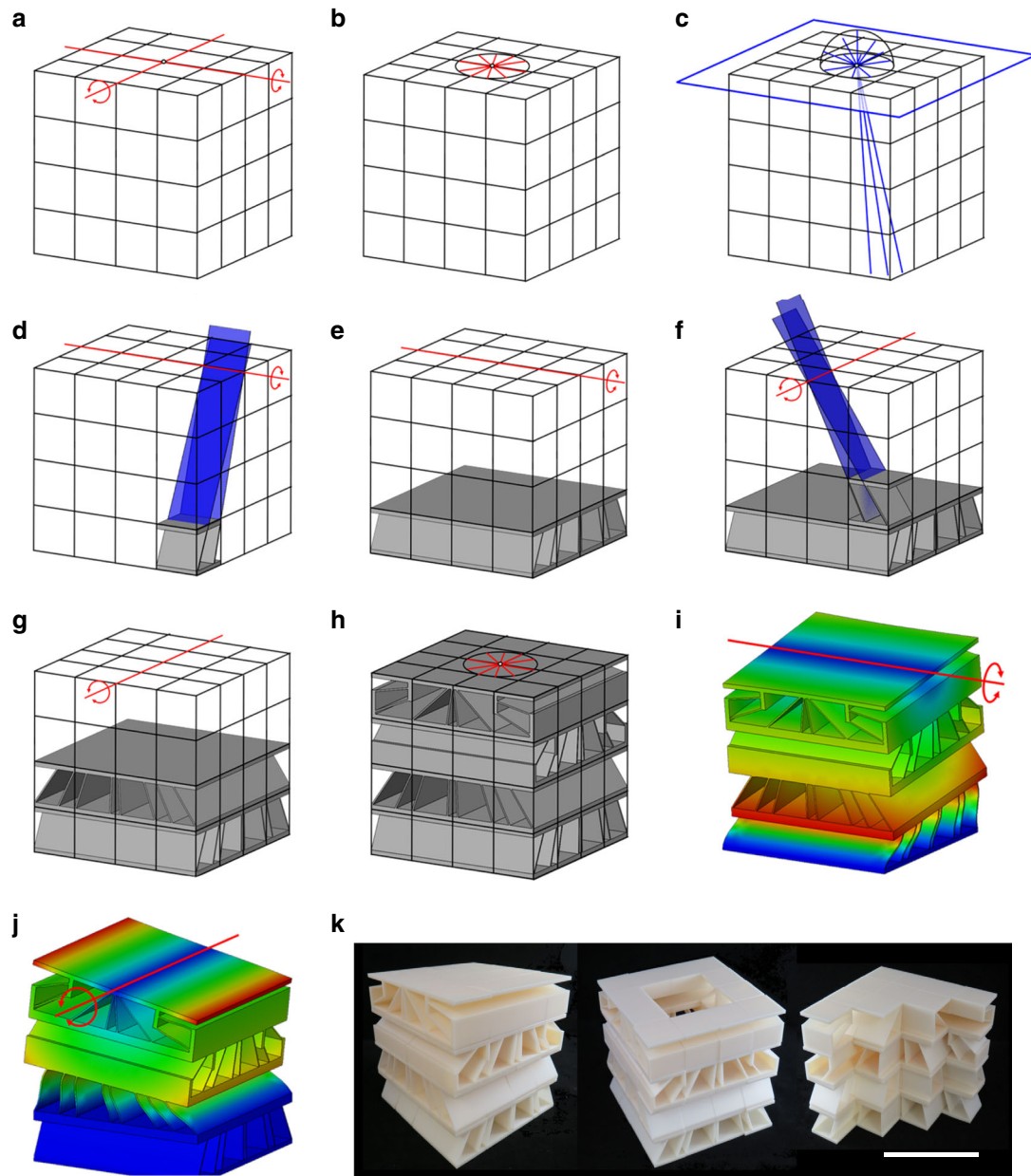

**Fig. 6** Example with two intersecting rotational degrees of freedom (DOFs). **a** Two desired intersecting rotational DOFs and, **b** their freedom and, **c** constraint spaces. Since the plane of the constraint space doesn't fill all space, two intermediate freedom spaces must be used. **d** The constraint space of the first intermediate freedom space is used to synthesize the unit cells of, **e** the first layer. **f** The constraint space of the second intermediate freedom space is used to synthesize the unit cells of, **g** the second layer. **h** This process is repeated for alternating layers until the final directionally compliant metamaterial (DCM) is synthesized that, **i**, **j** achieves the desired DOFs. **k** The DCM can be additively fabricated and shaped as desired (scale bar in **k**, 10 cm, and colors in **i** and **j** are defined in Fig. 3b)

labeled 2 DOF Type 8 in Supplementary Fig. 2 was chosen twice and oriented as shown in Fig. 7c. Note that these spaces do link to cell spaces and they combine to form the freedom space of Fig. 7b. The tool then uses the constraint spaces (Fig. 7d) of these intermediate freedom spaces to generate the appropriate number, location, and orientation of wire elements within each cell of the DCM (Fig. 7e, f). The tool also automatically generates layer extensions when necessary. Note from Fig. 7e, f that the wires within the alternating layers, labeled *L1* and *L2*, lie within the parallel disks of their respective constraint spaces and some of the cells required layer extensions. The tool then generates an .stl file of the resulting design (Fig. 7g), which can be uploaded to 3D printers (Fig. 7h). The tool also uses a custom-developed modal-analysis approach, which is provided in Supplementary Software and discussed in Methods, to generate animated .gif files of the DCM's DOFs (Fig. 7i). A demo of the tool is provided in Supplementary Movie 4. The computational times required by a standard desktop computer to generate *uxuxu* DCM designs that achieve the DOFs of Fig. 7a, i are plotted in Fig. 7j.

Although experienced engineers may be able to intuit some of the DCM designs provided previously, the automated tool of this work can rapidly generate designs that are too complex for most humans to visualize. Two such examples, which were generated by the tool, are provided in Fig. 8. The design of Fig. 8a achieves

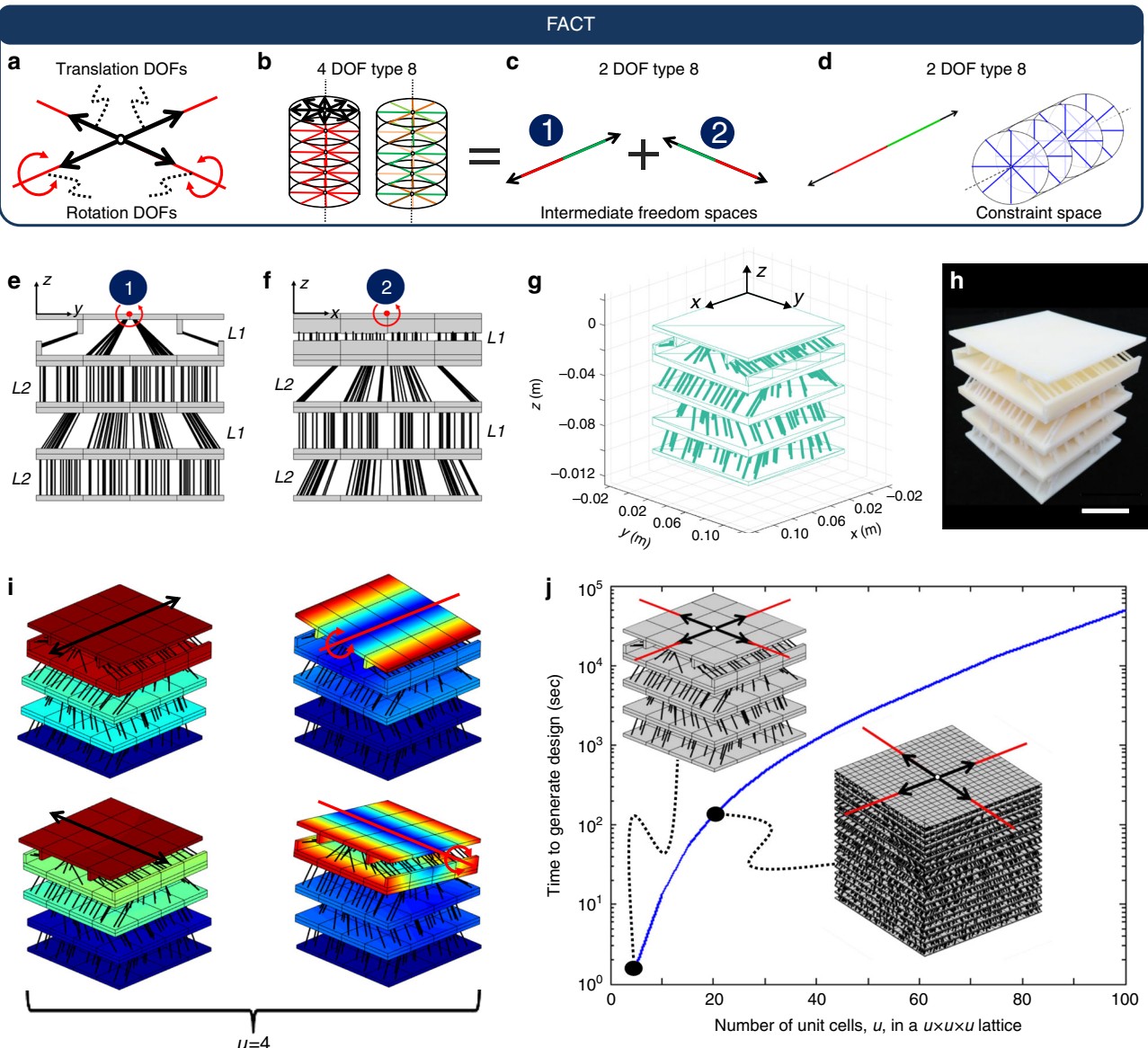

**Fig. 7** Automated design tool. **a** Users specify the desired degrees of freedom (DOFs) and, **b** the freedom space that results from the combination of those DOFs. If that freedom space does not link to a constraint space that is a cell space, **c**, intermediate freedom spaces must be selected from within the freedom space that do link to cell spaces. The tool then generates the directionally compliant metamaterial (DCM) topology using, **d** the constraint spaces of the intermediate freedom spaces. **e**, **f** The elements in the alternating layers lie within their corresponding constraint spaces. **g** The tool generates an.stl file that, **h** can be used to additively fabricate the design. **i** Animated .gif files of the DCM's DOFs are also generated. **j** Plot of the computational time required by the tool to generate a $u$x$u$x$u$ DCM that achieves the four DOFs specified (scale bar in **h**, 5 cm, and colors in **i** are defined in Fig. 3b)

the 2 DOF Type 4 freedom space in Supplementary Fig. 2. This freedom space consists of a disk of intersecting screws of the same pitch. The tool's modal analysis shows that the two independent screw DOFs (Fig. 8b, c) that combine to generate the desired freedom space are successfully achieved by the design generated. The design of Fig. 8d achieves the 3 DOF Type 6 freedom space in Supplementary Fig. 2. This freedom space consists of two parallel planes of parallel rotation lines oriented in orthogonal directions with respect to each other and a translation arrow that is perpendicular to these planes. The freedom space also possesses other screw lines that are not shown in Fig. 8d to avoid visual clutter. The tool's modal analysis shows that the two desired independent rotational DOFs and the one desired independent translational DOF (Fig. 8e–g) that combine to generate the desired freedom space are successfully achieved by the design

generated. Animated .gif files that show how the designs of Fig. 8 deform are provided in Supplementary Movie 4.

**Effect of bulk shape on DOFs**. The DOFs achieved by a DCM are similarly affected by its bulk shape and architecture. Thus, the freedom space of a DCM is determined by linearly combining the twist vectors that constitute the freedom space of the DCM's architecture and the freedom space of the DCM's bulk shape if it were filled with a homogenous material. We experimentally demonstrate this principle using the example of Fig. 9 (see Supplementary Movie 5). The freedom space of a homogenous material shaped like the system shown in Fig. 9a is a single translation arrow (i.e., 1 DOF Type 3 in Supplementary Fig. 2). If the same shape rotated 90° is used as the system's

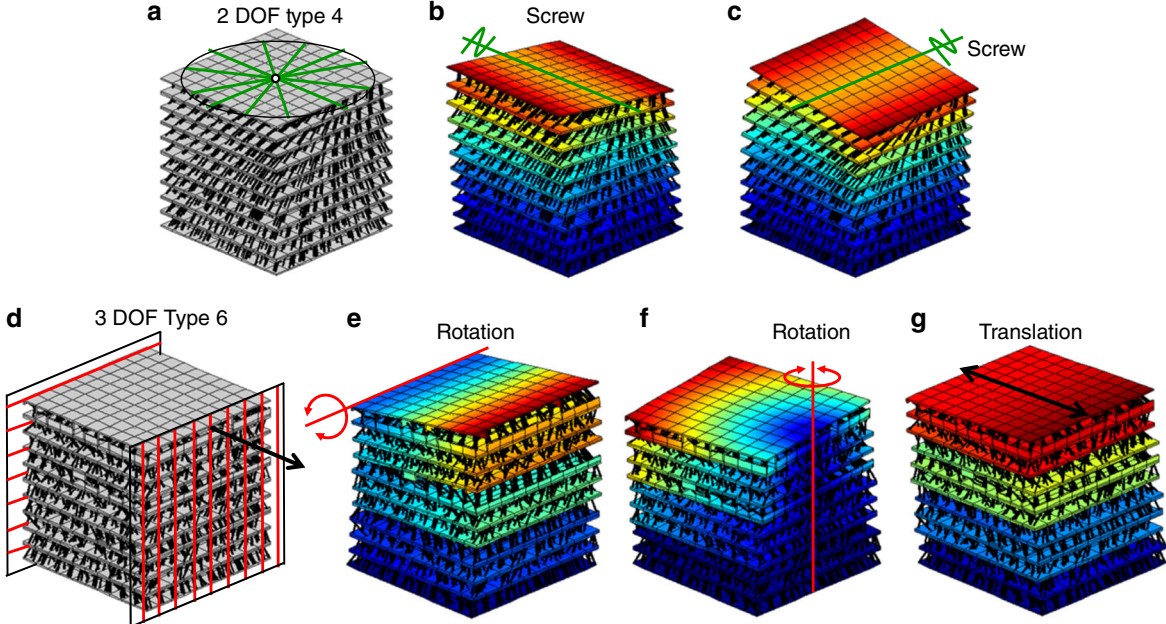

**Fig. 8** Less intuitive designs generated by the automated tool. **a** A design that achieves, **b**, **c** two intersecting screw degrees of freedom (DOFs) with the same pitch. **d** A different design that achieves, **e**, **f** two orthogonally skew rotational DOFs and, **g** an orthogonal translational DOF

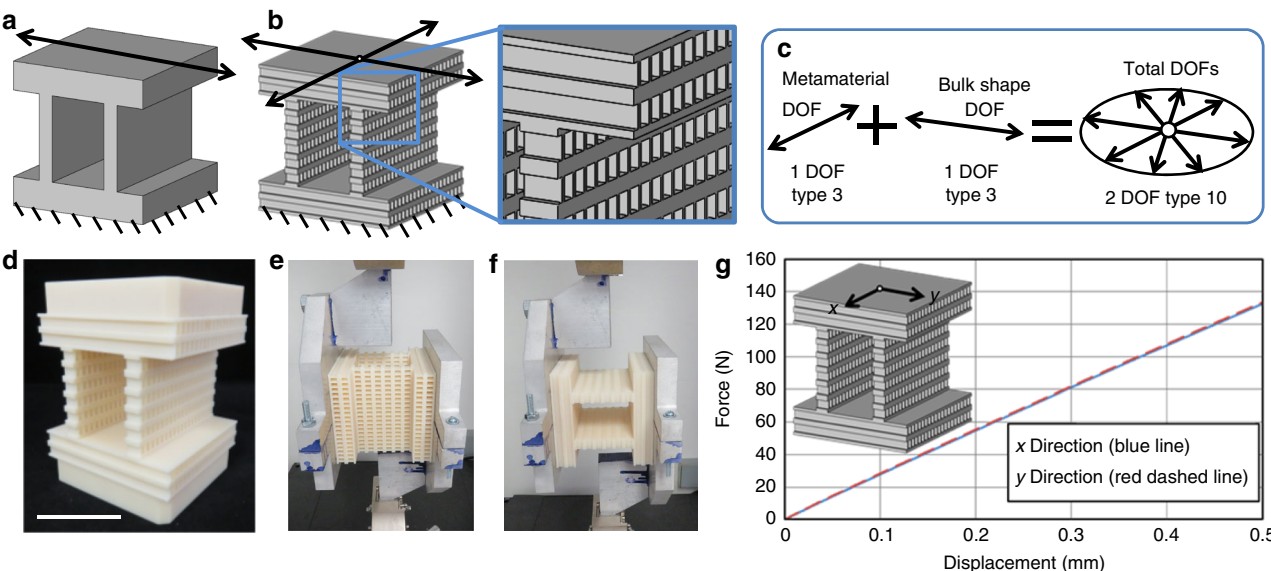

**Fig. 9** Directionally compliant metamaterial (DCM) shape affects degrees of freedom (DOFs). **a** A bulk shape that achieves a translational DOF but would, **b** achieve two translational DOFs if the shape were rotated 90° and used inside of itself. **c** The freedom space of a DCM results from the sum of its architecture's freedom space and its bulk shape's freedom space. **d** DCM was additively fabricated and, **e**, **f** tested in two different directions. **g** The stiffness in both directions are the same (scale bar in **d**, 5 cm)

periodic architecture as shown in Fig. 9b, the freedom space of the resulting DCM, as predicted by the principle discussed previously, is the disk of translations (2 DOF Type 10) shown in Fig. 9c. This freedom space results from the linear combination of the translational DOF of the DCM's bulk shape and the translational DOF of the DCM's architecture. The DCM was 3D printed (Fig. 9d) and loaded along the two directions of the DCM's translational DOFs (Fig. 9e, f). The plot of Fig. 9g demonstrates that the compliance along these directions (i.e., $x$ and $y$-axes) are similar. Another example that demonstrates this section's principle is provided in Supplementary Note 2

and shown in Supplementary Fig. 4 and in Supplementary Movie 5.

## Discussion

We created an approach that leverages the vector spaces of the FACT library to enable the automated synthesis of metamaterials (i.e., DCMs) that achieve desired combinations of compliant DOFs while assuming any form. The reason such materials can achieve these properties is that their DOFs are independently determined by both the DOFs of their architecture and the DOFs

of their bulk shape. To maintain this independence for ensuring high shape versatility, the cell resolution of DCMs should be sufficiently high (i.e., cell size should be orders of magnitude smaller than the characteristic size of the DCM) such that enough redundant cells exist in the architecture to span any cross-section of the material's shape.

Compared to other computational approaches (e.g., topology optimization) that typically require 10 s of hours to generate a single two-dimensional (2D) cell design within a periodic meta-material, the approach proposed here requires only 10 s of seconds to generate thousands of different 3D cell designs within aperiodic DCMs (i.e., ~6 orders of magnitude more cells per second can be generated). It also does not require pre-computed databases of cell designs[5,6], which typically demand significant time to populate and large amounts of memory to store. Rather, the approach rapidly searches the most promising branches of the mathematically complete design tree to generate DCM solutions, which enable irregularly shaped flexure bearings, compliant prosthetics, morphing structures, and soft-robots that are too complex to synthesize using alternative approaches. The theory introduced also paves the way for enabling the synthesis of general metamaterial configurations beyond the stacked-layer serial designs of this work.

## Methods

**FACT library**. Supplementary Fig. 2 contains the mathematically complete library of all 50 freedom spaces. The chart organizes the freedom spaces into seven different columns according to the number of DOFs that combine to generate them. Each freedom space is labeled with a type number at its upper left corner. The freedom spaces that lie outside the black-outlined pyramid of Supplementary Fig. 2 are not shown with their complementary constraint spaces because such spaces do not possess enough independent PFWVs to synthesize parallel flexure systems[15–17] (i.e., systems that directly join two bodies together using parallel elements like the layered cells within the DCMs of this paper). Additionally, only freedom spaces that link to constraint spaces that are cell spaces (i.e., spaces that lie within the region shaded yellow in Supplementary Fig. 2) can fill any volume of space with enough independent PFWVs to enable the synthesis of cells that successfully achieve their intended DOFs regardless of where they are located in a DCM. Although others have mathematically categorized screw systems similar to the vector spaces of FACT for other applications[21–26], the library of Supplementary Fig. 2 has been organized to facilitate the design of DCMs. Exploded views of each freedom and constraint space type in the library are provided and described in detail with the equations that define their geometry in prior publications[15,16].

**Mathematically defining PFWVs**. PFWVs[15–20], $\mathbf{W}_{6\times1}$, are graphically depicted as the blue constraint-force lines (Supplementary Fig. 5) that constitute the constraint spaces of the FACT library. These vectors are defined according to

$$\mathbf{W}_{6\times1} = [\,\mathbf{f}_{1\times3} \quad \mathbf{r}_{1\times3} \times \mathbf{f}_{1\times3}\,]^T \tag{1}$$

where $\mathbf{f}_{1\times3}$ is a $1\times3$ force vector that points in the direction of the blue constraint-force line's axis, and $\mathbf{r}_{1\times3}$ is a $1\times3$ location vector that points from the coordinate system to any location along that line's axis. Physically speaking, blue constraint-force lines represent the axis about which a force can be imparted.

**Modeling general flexible element geometries**. There are three categories of flexible elements—parallel[27], serial[28], and hybrid[29]. Since parallel elements are sufficient for generating DCM examples that achieve any desired combination of DOFs and require the least amount of computation to model them compared to serial or hybrid elements, parallel elements are used exclusively to generate the DCMs of this work. If, however, a future work desires serial or hybrid elements, the theory to model them exists[28,29].

An element is parallel if blue constraint-force lines can fill the element's entire geometry without exiting the geometry at any point and directly connect the two rigid bodies that the element joins together. A parallel element is modeled using the constraint space that graphically depicts the linear combination of the constraint-force lines' corresponding PFWVs that satisfy the previous two conditions. Thus, the constraint space of an element represents the forces that the element is capable of resisting (i.e., the element's directions of highest stiffness). As an example, consider the parallel wire element shown in Supplementary Fig. 5. The constraint space that models this element is the single blue constraint-force line that satisfies the two conditions specified above. This model treats the wire element as if it is infinitely stiff along its axis but is infinitely compliant in all other directions since the constraint-force line can only impart forces along its axis. Additionally, note

that the constraint space models only the location and orientation of the element and does not consider its material properties or its geometric parameters (i.e., its diameter or length).

All other parallel element geometries can be similarly modeled. Example parallel elements and the constraint spaces that model their behavior are shown in Supplementary Fig. 6. The DOF column and type numbers for each of these constraint spaces are labeled in the figure using the convention established in Supplementary Fig. 2.

**Modeling DOFs and freedom spaces**. Just as constraint spaces are generated by linearly combining their independent PFWVs defined in equation (1), freedom spaces are generated by linearly combining their DOFs. DOFs can be mathematically modeled using $6\times1$ twist vectors[15–20], $\mathbf{T}_{6\times1}$, defined by

$$\mathbf{T}_{6\times1} = [\,\boldsymbol{\omega}_{1\times3} \quad (\mathbf{c}_{1\times3} \times \boldsymbol{\omega}_{1\times3}) + p\boldsymbol{\omega}_{1\times3}\,]^T \tag{2}$$

where $\boldsymbol{\omega}_{1\times3}$ is a $1\times3$ angular velocity vector that points along the twist's axis, $\mathbf{c}_{1\times3}$ is a $1\times3$ location vector that points from the coordinate system to any location along the twist's axis, and $p$ is the scalar pitch of the twist. If the twist's pitch is zero, the twist is a red rotation line. If the twist's pitch is any other finite nonzero value, the twist is a green screw line. If the twist's pitch is infinity, the twist is a black translation arrow and is defined according to

$$\mathbf{T}_{6\times1} = [\,\mathbf{0}_{1\times3} \quad \mathbf{v}_{1\times3}\,]^T \tag{3}$$

where $\mathbf{0}_{1\times3}$ is a $1\times3$ zero vector, and $\mathbf{v}_{1\times3}$ is a $1\times3$ linear velocity vector that points along the axis of the twist. Although all the compliant directions contained within a freedom space are modeled using twist vectors, the DOFs of a freedom space are the independent twist vectors that linearly combine to generate the other twist vectors (i.e., compliant directions) within the freedom space.

**Selecting elements within constraint spaces**. This section explains how constraint spaces can be used to determine the location and orientation of flexible elements from within the constraint spaces' geometries to ensure that the resulting system achieves its intended DOFs. For a parallel system to successfully achieve the $n$ DOFs of its intended freedom space, flexible elements that collectively contain $m$ independent PFWVs should be selected from within the freedom space's complementary constraint space where

$$m = 6 - n \tag{4}$$

Thus, since the freedom space of Fig. 2b consists of one screw DOF (i.e., $n = 1$), each cell within the final DCM (Fig. 2c) requires flexible elements that together contain $m = 6-n = 5$ independent PFWVs from within the freedom space's complementary constraint space. Consequently, each of the cells in the DCM of Fig. 2c consist of five wire elements with axes that are colinear to five independent PFWVs from within the constraint space of Fig. 2b.

Thus, although Eq. (4) can be used to determine the correct number, $m$, of independent PFWVs to select from within a constraint space, the equation does not provide guidance on how to select the $m$ PFWVs such that they are independent. Gaussian elimination[30] could be used as a mathematical approach to confirm whether a collection of $m$ PFWVs are independent by determining if a matrix consisting of the PFWVs possesses a rank of $m$. The rules provided with the shapes of Supplementary Fig. 7, however, offer a more intuitive approach for selecting PFWVs from constraint spaces such that they are independent. Each constraint space in the FACT library consists of various combinations of the nine shapes shown in Supplementary Fig. 7a-i. The instructions above each shape in the figure describe how many independent PFWVs lie within the shape and how they should be selected from the shape such that they will be independent.

Different flexible elements contain different numbers of independent PFWVs within their geometry. Whereas a wire element contains a single independent PFWV, blade elements contain three independent PFWVs. The number of independent PFWVs within a general flexible element is the number of independent PFWVs within the element's constraint space. Thus, the number, $m$, of independent PFWVs within each element shown in Supplementary Fig. 6 can be determined by subtracting the labeled DOF number, $n$, from 6 according to Eq. (4).

The principles of this section can be used to synthesize the parallel topologies of general DCM cells. Suppose, as an example, a parallel cell is desired that achieves a single rotational DOF located on the edge of the cell's two rigid bodies as shown in Supplementary Fig. 8a. The complementary constraint space of this single-rotation freedom space, labeled 1 DOF Type 1 in Supplementary Fig. 2 and shown larger in Supplementary Fig. 8b, is the set of planes that intersect the rotation's axis. Thus, according to Eq. (4), $m = 5$ total independent PFWVs must be selected from within this constraint space because its freedom space contains $n = 1$ DOF. Since the constraint space consists of intersecting planes, which according to Supplementary Fig. 6 are each the constraint space of a single blade element (i.e., 3 DOF Type 1), a blade element could be selected from within any one of the intersecting planes. Additionally, since the planar constraint space of a blade element contains only three independent PFWVs according to Supplementary Fig. 7d, two more

independent PFWVs must be selected from within the constraint space of Supplementary Fig. 8b to ensure that the resulting cell achieves the desired rotational DOF only. Thus, two wire elements could be selected with axes that are colinear with PFWVs that lie within another plane in the constraint space as shown in Supplementary Fig. 8a. A different view of the same cell is shown in Supplementary Fig. 8c. The resulting cell would be stiff in all directions except about the desired rotational DOF.

The cell in Supplementary Fig. 8a,c is exactly-constrained[16,31] because the sum of the independent PFWVs contained within each of its elements equals $m$ from Eq. (4) (i.e., three independent PFWVs from the blade added to one independent PFWV from each of the two wires equals five independent PFWVs, which is how many independent PFWVs lie within the cell's constraint space). If additional elements had been selected from within the planes of the constraint space beyond the blade and two wire elements, the resulting cell would possess redundant elements and would be over-constrained[16,31]. Note all the cell designs that constitute the DCMs of Figs. 1b, 4e, 6h, and 9b are over-constrained whereas all the cell designs that constitute the DCMs of Figs. 2c, 5c, 7j, 8a, and 8d are exactly-constrained. Over-constraining the cells within DCMs has little effect on their overall behavior since DCM layers are already heavily over-constrained by the redundant cells that constitute their layers. This cell redundancy is at the core of why DCM's can be shaped in arbitrary ways without compromising their desired DOFs. Thus, although it is acceptable to over-constrain individual cells, it is typically computationally more efficient to synthesize exactly-constrained cells because such cells possess the fewest number of necessary elements.

As another example, exactly-constrained DCM cells consisting of different flexible elements could be synthesized to achieve the screw DOF of Fig. 2b. Consider the cell, shown from two different views in Supplementary Fig. 8d, e, that achieves a single screw DOF (i.e., $n = 1$) because it contains two wire elements with axes that are colinear with two independent PFWVs within the constraint space of Fig. 2b and one circular-hyperboloid element, labeled 3 DOF Type 7 in Supplementary Fig. 6, which contains three independent PFWVs that also lie within the constraint space. Thus, because the sum of the independent PFWVs contained within each of its elements equals $m$ from Eq. (4) (i.e., three independent PFWVs from the circular-hyperboloid element added to one independent PFWV from each of the two wires equals five independent PFWVs, which is how many independent PFWVs lie within the cell's constraint space since $n = 1$).

**Selecting intermediate freedom spaces within freedom spaces**. If a DCM's desired freedom space does not link to a constraint space that is a cell space, intermediate freedom spaces must be selected that do link to constraint spaces that are cell spaces to successfully synthesize the DCM. The twist vectors within the intermediate freedom spaces selected must linearly combine to generate the twist vectors within the desired freedom space only. Any number of intermediate freedom spaces can be selected, but each selected intermediate freedom space represents the DOFs achieved by its corresponding alternating layer within the DCM. Thus, if $L$ intermediate freedom spaces are selected, the DCM needs to possess at least $L$ layers to successfully achieve the desired freedom space.

**FEA details**. Abaqus was used to perform the FEA on the DCM in Fig. 3b using 10-node quadratic tetrahedral elements (C3D10). A linear elastic material model was used with the parameters of Nanoscribe Ip-Dip polymer. Specifically, a Young's Modulus of 2.7 GPa and a Poisson's ratio of 0.49 were used. A total of 940,000 elements were used to ensure mesh convergence. A vertical displacement was applied to the nodes on the pyramid, while their in-plane DOFs were left unconstrained. The DOFs of the nodes at the base of the DCM were fully constrained. During compression, the DOFs from the nodes at each layer's corners were used to calculate the corresponding rotations. SolidWorks was used to perform the linear modal analyses for the case studies of Figs. 4g–i, 5d–f, 6i, j, and Supplementary Fig. 4d-f using the default mesh settings. Although almost any constituent material could have been used to produce the first $n$ mode shapes such that they correspond with the FACT predicted $n$ DOFs for any of these case studies, the default properties of Acrylonitrile Butadiene Styrene (ABS) where used to generate the results in the figures.

**Mathematics underlying the automated tool**. The mathematics of this section enabled the automated design tool provided in Supplementary Software. If a DCM cell's topology is to be synthesized such that it achieves a certain freedom space, the constraint space of that freedom space can be calculated according to

$$[\mathbf{T}_{FS}]_{n \times 6}[\mathbf{\Delta}]_{6 \times 6}\mathbf{W}_{6 \times 1} = \mathbf{0}_{n \times 1} \tag{5}$$

where $[\mathbf{T}_{FS}]_{nx6}$ is an $n$ x 6 matrix that contains the transpose of the freedom space's, $n$, independent DOF twist vectors arranged in $n$ rows, and $[\mathbf{\Delta}]_{6 \times 6}$ is a $6 \times 6$ matrix defined by

$$[\mathbf{\Delta}]_{6 \times 6} = \begin{bmatrix} [\mathbf{0}]_{3 \times 3} & [\mathbf{I}]_{3 \times 3} \\ [\mathbf{I}]_{3 \times 3} & [\mathbf{0}]_{3 \times 3} \end{bmatrix} \tag{6}$$

where $[\mathbf{0}]_{3 \times 3}$ is a $3 \times 3$ matrix filled with zeros and $[\mathbf{I}]_{3 \times 3}$ is a $3 \times 3$ identity matrix. Note from Eq. (5) that the null space of $[\mathbf{T}_{FS}]_{nx6}[\mathbf{\Delta}]_{6 \times 6}$ is the linear combination of $m$ independent $6 \times 1$ wrench vectors, $\mathbf{W}_{6 \times 1}$. The freedom space's complementary constraint space geometrically represents this linear combination, which can be mathematically modeled using a $6 \times m$ matrix, $[\mathbf{W}_{CS}]_{6 \times m}$, that consists of the independent wrench vectors, $\mathbf{W}_{6 \times 1}$, arranged in columns according to

$$[\mathbf{W}_{CS}]_{6 \times m} = \begin{bmatrix} W_{1,1} & W_{1,2} & W_{1,3} & W_{1,4} & W_{1,5} & W_{1,6} \\ W_{2,1} & W_{2,2} & W_{2,3} & W_{2,4} & W_{2,5} & W_{2,6} \\ \vdots & \vdots & \vdots & \vdots & \vdots & \vdots \\ W_{m,1} & W_{m,2} & W_{m,3} & W_{m,4} & W_{m,5} & W_{m,6} \end{bmatrix}^{T} \tag{7}$$

where $W_{i,j}$ is the $j$th component in wrench vector, $i$. Recall that the relationship between $m$ and $n$ is given in Eq. (4). If $\mathbf{A}_{m \times 1}$ is a $m$x1 vector where each of its $m$ components can be any real number, then $[\mathbf{W}_{CS}]_{6 \times m}\mathbf{A}_{m \times 1}$ represents any wrench vector within the constraint space. Not all of the wrench vectors in the constraint space are guaranteed to be PFWVs, i.e., wrench vectors of the form given in Eq. (1) where the force vector, $\mathbf{f}_{1 \times 3}$, consists of three components, $f_1$, $f_2$, and $f_3$, according to

$$\mathbf{f}_{1 \times 3} = [f_1 \quad f_2 \quad f_3] \tag{8}$$

and where the location vector, $\mathbf{r}_{1 \times 3}$, possesses three components, $r_1$, $r_2$, and $r_3$, according to

$$\mathbf{r}_{1 \times 3} = [r_1 \quad r_2 \quad r_3] \tag{9}$$

Since it is necessary to identify $m$ independent PFWVs that lie within the constraint space of Eq. (7) and directly join the cell's two rigid bodies together to correctly place flexible elements in the DCM's cell, a location vector, $\mathbf{r}_{1 \times 3}$, for one of these elements is first selected. This location vector points from the coordinate system to a random point generated between specific bounds on the bottom side of cell (a)'s upper rigid body, labeled in Supplementary Fig. 9. If the wire element being placed possesses a radius of $R$, its location vector's $x$-axis component, $r_1$, should be greater than or equal to $x_a + R$ and less than or equal to $x_a + s - R$ so that the element doesn't spill into the space designated for the neighboring cell. Note from Supplementary Fig. 9 that $s$ is the side length of each cube-shaped cell. Similarly, the vector's $y$-axis component, $r_2$, should be greater than or equal to $y_a + R$ and less than or equal to $y_a + s - R$ for the same reason. The vector's $z$-axis component, $r_3$, should equal $z_a + t$, where $t$ is the thickness of the cell's two rigid bodies. Given the random location vector selected within these bounds on the bottom side of the cell's upper rigid body, the equation

$$[f_1 \quad f_2 \quad f_3 \quad (r_2 f_3 - r_3 f_2) \quad (r_3 f_1 - r_1 f_3) \quad (r_1 f_2 - r_2 f_1)]^{T} = [\mathbf{W}_{CS}]_{6 \times m}\mathbf{A}_{m \times 1} \tag{10}$$

is enforced to ensure that wrench vectors are identified from within the constraint space that pass through the random point, $\mathbf{r}_{1 \times 3}$, but are also PFWVs according to the form in Eq. (1). By substituting the top three rows of Eq. (10) into the bottom three rows of the same equation, another equation is derived according to

$$[\mathbf{M}]_{3 \times m}\mathbf{A}_{m \times 1} = \mathbf{0}_{m \times 1} \tag{11}$$

where

$$[\mathbf{M}]_{3 \times m} = \begin{bmatrix} (r_2 W_{1,3} - r_3 W_{1,2} - W_{1,4}) & (r_2 W_{2,3} - r_3 W_{2,2} - W_{2,4}) & \cdots & (r_2 W_{m,3} - r_3 W_{m,2} - W_{m,4}) \\ (r_3 W_{1,1} - r_1 W_{1,3} - W_{1,5}) & (r_3 W_{2,1} - r_1 W_{2,3} - W_{2,5}) & \cdots & (r_3 W_{m,1} - r_1 W_{m,3} - W_{m,5}) \\ (r_1 W_{1,2} - r_2 W_{1,1} - W_{1,6}) & (r_1 W_{2,2} - r_2 W_{2,1} - W_{2,6}) & \cdots & (r_1 W_{m,2} - r_2 W_{m,1} - W_{m,6}) \end{bmatrix} \tag{12}$$

and $\mathbf{0}_{mx1}$ is a $m$ x 1 zero vector. If the $o$ independent vectors that result from the null space, $\mathbf{A}_{mx1}$, of $[\mathbf{M}]_{3 \times m}$ are arranged within an $m$ x $o$ matrix, $[\mathbf{A}]_{mxo}$, according to

$$[\mathbf{A}]_{m \times o} = \begin{bmatrix} A_{1,1} & A_{2,1} & \cdots & A_{o,1} \\ A_{1,2} & A_{2,2} & \cdots & A_{o,2} \\ \vdots & \vdots & \ddots & \vdots \\ A_{1,m} & A_{2,m} & \cdots & A_{o,m} \end{bmatrix} \tag{13}$$

the force vector, $\mathbf{f}_{1 \times 3}$, of all the PFWVs that lie in the constraint space and pass through the point, $\mathbf{r}_{1 \times 3}$, can be determined according to

$$\mathbf{f}_{1 \times 3} = ([\mathbf{W}_{CSsub}]_{3 \times m}[\mathbf{A}]_{m \times o}\mathbf{a}_{o \times 1})^{T} \tag{14}$$

where

$$[\mathbf{W}_{\text{CSsub}}]_{3\times m} = \begin{bmatrix} W_{1,1} & W_{2,1} & \cdots & W_{m,1} \\ W_{1,2} & W_{2,2} & \cdots & W_{m,2} \\ W_{1,3} & W_{2,3} & \cdots & W_{m,3} \end{bmatrix} \qquad (15)$$

and $\mathbf{a}_{o\times1}$ is an $o$ x 1 vector with $o$ components of any real and finite value. Since only one PFWV that lies within the constraint space and passes through the point, $\mathbf{r}_{1\times3}$, is necessary to place a wire element, the automated approach randomly assigns the components within $\mathbf{a}_{o\times1}$ to be any value between $-1$ and 1. The next step is to check if the resulting PFWV generated also directly joints the two rigid bodies together within the cell. To this end, the conditions $(x_a + R) \leq (r_1 + f_1 b) \leq (x_a + s - R)$ and $(y_a + R) \leq (r_2 + f_2 b) \leq (y_a + s - R)$ are enforced, where $b$ can be solved using $r_3 + f_3 b = z_a + s - t$. Thus, the final conditions enforced are

$$(x_a + R) \leq \left(r_1 + \frac{f_1}{f_3}(s - 2t)\right) \leq (x_a + s - R) \qquad (16)$$

and

$$(y_a + R) \leq \left(r_2 + \frac{f_2}{f_3}(s - 2t)\right) \leq (y_a + s - R) \qquad (17)$$

If the PFWV generated satisfies these conditions, a wire element with a radius of $R$ joins the cell's bodies together starting from the point $\mathbf{r}_{1\times3}$ and ending where the wire passes through the top surface of the cell's lower rigid body along the vector, $\mathbf{f}_{1\times3}$, which defines the wire's axis. This approach is repeated until each cell within the entire DCM is exactly-constrained by $m$ wire elements that are colinear with independent PFVWs that satisfy the conditions discussed above. An algorithm is also provided in Supplementary Software for including layer extensions where necessary.

**Custom-developed modal analysis approach**. Embedded within the automated tool of Supplementary Software is a simplified modal analysis approach that enables the DOF verification and animation of the DCMs that the tool designs. Most DCMs require this simplified approach to analyze their DOFs because traditional FEA packages become overwhelmed by the extreme number of elements that constitute DCM architectures. The simplified approach was used to generate Fig. 7i, j, and Fig. 8a–g. The approach constructs a specialized stiffness matrix[32], [**Stiff**], by treating each wire or blade within the DCM being analyzed as a single beam element. A specialized mass matrix[33], [**Mass**], is also constructed using the mass and mass moments of inertia about the centers of mass of each of the DCM's rigid layers. The eigen values of $[\mathbf{Mass}]^{-1}[\mathbf{Stiff}]$ are then calculate to determine and animate the first $n$ mode shapes, which will typically correspond directly to the DCM's intended $n$ DOFs (or at very least they will correspond to the linear combinations of these DOFs).

**Code availability**. The Supplementary Software code is available using a GitHub repository link provided below. Additional code used to generate the plots in the paper beyond that found in Supplementary Software are available from the corresponding author upon request. (https://github.com/jonathanbhopkins/Computationally-Efficient-Design-of-Directionally-Compliant-Metamaterials.git)

## Data availability
The authors declare that all data supporting the findings of this study are included in the main manuscript file or Supplementary Information or are available from the corresponding author upon request. The computer-aided design (CAD) models necessary to replicate the FEA results of this study are also available from the corresponding author upon request.

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

## Acknowledgements
This work was supported by AFOSR under award number FA9550-15-1-0321. J.B.H. acknowledges program officer Byung "Les" Lee. Daryl Yee is also gratefully

acknowledged for his support fabricating the DCM of Fig. 3a. J.R.G. gratefully acknowledges financial support of the Department of Defense through Vannevar-Bush Faculty Fellowship.

## Author contributions

L.A.S. coded the GUI for the automated tool and generated the results in Fig. 7. F.S. made the parts and performed the study of Fig. 9. C.M.P. collected the data in Fig. 3 and generated the results of Supplementary Fig. 3. R.I.B. performed the FEA for the DCM of Fig. 3. J.R.G. managed C.M.P. and helped revise the manuscript. J.B.H. conceived the idea of DCMs, created the theory to synthesize them, generated the DCM examples, coded the enabling portion of the automation tool, wrote the paper, made the figures, and managed the project.

## Additional information

**Competing interests:** The authors declare no competing interests.

