## [Peer Review File · Nature Communications]

Reviewers' comments:

Reviewer #1 (Remarks to the Author):

The manuscript "Computationally efficient design of directionally compliant metamaterials" by Shaw et al. reports the design, fabrication and experimental validation of a novel class of metamaterials which exhibit prescribed deformations, such as torsion, rotation and shear. Their design strategy hinges on a purely kinematic design approach, which translates target motions into sets of constraints. In turn, such constraints can be expressed mathematically as a linear system of equations, that can be solved very rapidly numerically. As a result, the internal architecture of metamaterials with predefined macroscopic shapes (e.g. 5x5x5 cube) can be designed for a wide range of mechanical deformations, from motions with unique degrees of freedom to motions with multiple (up to 3) degrees of freedom. Although their design strategy is not new (it was already developed for compliant mechanisms), its generalisation in the context of metamaterials is novel and a very significant addition to the metamaterial toolbox. I believe that the results reported here will be of strong interest to mechanical engineers, physicists and material scientists interested in metamaterials. The off-the-self computational tool they provide might even make it readily useable beyond the scientific community, e.g. by product designers. Therefore I think that this manuscript is a potential very good fit for Nature Communications. That being said, I think that as it stands, the paper lacks too much precision and clarity to be fully convincing and to be readable by the broad readership of Nature Communications. For these reasons, I strongly encourage the authors to address the following comments:

1. The extensive use of acronyms (more than 1 per line on average) makes the paper extremely difficult to follow. I believe that a significant rephrasing would allow the authors to use less repetitions and therefore to no rely that much on acronyms.
2. In some instances, the vocabulary is not precise enough, e.g. in the definition of "the yellow shaded region..." (l 143, p7), what is a "viable topology"? Also, always referring to the labelled yellow region is extremely confusing because as a reader, we loose track of the underlying scientific nature of such class of freedom spaces.
3. In all the FEA plots (Fig. 3b, Figs.4g,h,i, figs. 5d,e,f, Figs. 6i,j, Fig. 7i, Figs. 8 b,c,e,f,g), it is never clearly stated in the caption which numerical protocol has been used to obtain these results. I understand from the text description of Fig. 4 that modal analysis has been used, but what about for the other figures? In general, the description of the FEA is too scarce to assess their validity and to reproduce them, e.g. the description of boundary conditions, mesh size, constitutive models are absent.
4. The description of the metamaterial geometric feature is incomplete: how are the beams and blades thickness chosen? How does this choice determine the compliance for target compliant deformations and for target non-compliant deformations.
5. In general I feel that despite the generic design approach for the nature (wire, blade, etc..) and orientation of the slender elements, many design choices remain arbitrary and poorly explained. For instance, In fig. 2.: what sets the choice of five wires per unit cell? In fig. 4.: what sets the choice of two blades per unit cell? in fig 4, from cell to cell, the blades are aligned in the vertical direction but not in the horizontal direction: why such a difference?
6. To convincingly demonstrate that the metamaterials are directionally compliant, it would be appropriate to provide measurements of the non-compliant deformation modes, e.g. in figure 9.
7. About the counting of constraints and degrees of freedom in formula (4) of the methods: it is well known that the effects of multiple constraints can be redundant if they are related by symmetries, for instance when multiple bars are aligned (See Calladine, IJSS 1978). How do the

authors treat such issue?

8. So far, the design approach has exclusively focussed on kinematics. However, I believe that the approach of the authors could easily be extended using e.g. beam and plate theory to also determine/design for a target compliance. This is probably outside of the scope of this work, but I am curious to hear the authors' take on this and I think that this aspect would be worth mentioning. □

9. A related question concerning fig 9: I understand that the authors designed for the X and Y direction to be compliant, but they report very similar values of compliances in the X and Y directions. I find this result striking. Did the authors design for matching the X and Y compliances or is this just a coincidence? If it has been designed, the authors should explain how.

10. Could the authors comment of the importance of the nonlinear response? Since their structures comprise many slender elements, I would expect strong geometric nonlinearities to play a role.

11. What determines the optimal choice for the number of unit cells?

Reviewer #2 (Remarks to the Author):

A new computational tool was created to design directionally compliant metamaterials more efficiently, since design of metamaterials is complex due to the number and locations of flexible elements. The new software tool takes orders of magnitude less time (tens of seconds vs tens of hours) than current computational tools for this application.

This computational model was based on the Freedom and Constraint Topologies (FACT) methodology with some simplified assumptions in order to more efficiently compute designs. The simplified assumptions are that only geometry and orientation are considered in designing and the beams are infinitely stiff along the constraint-force lines but compliant in all other directions.

Basic rules for use of the FACT methodology and mathematical definitions to model the methodology within the tool were presented to explain how the tool determines the design. Case studies were performed on single degree-of-freedom (DOF), multi-DOF, multi-DOF with CS outside the yellow shaded region, and bulk shape systems to show the design works for many different cases.

The discussion on the FACT methodology used in each case studies along with the accompanying figures clearly shows the process by which the computational tool efficiently designs these compliant metamaterials. This helped me to better understand how the methodology was applied to design the computational tool.

There is an impressive variety of case studies presented. This showed that the tool has been tested and can be used for many different designs. The supplementary information added even more case studies to show that this tool works for many different cases.

CHANGES THAT SHOULD BE MADE BEFORE PUBLICATION

At Line 435, "existing" should be "exiting".

CHANGES THAT SHOULD BE CONSIDERED TO IMPROVE THE PAPER

The flow and readability of Figure 2 could be improved by placing 2d, 2e prior to 2c. This would give the figure a more linear feel as to how the process works.

Suggest placing the comments about scale bars and color definitions in the corresponding subfigure description in Figures 3, 6, 7, and 9. For example, for Figure 3 – Place “(scale bar in a, 50 μm)” at the end of the description of a. As another example, for Figure 6 – Place “colors in i and j are defined in Fig. 3b” at the end of the description of i, j.

Lines 165 – 170 discuss that layer extensions may be added to the design. Is there any time where these extensions could interact with a different layer and would thus affect the desired compliance of the design? In looking at this in the context of the rest of the paper I don't think there would be interactions but discussing that the layer extensions won't affect the desired compliance may be helpful.

Reviewer #1:

The manuscript “Computationally efficient design of directionally compliant metamaterials” by Shaw et al. reports the design, fabrication and experimental validation of a novel class of metamaterials which exhibit prescribed deformations, such as torsion, rotation and shear. Their design strategy hinges on a purely kinematic design approach, which translates target motions into sets of constraints. In turn, such constraints can be expressed mathematically as a linear system of equations, that can be solved very rapidly numerically. As a result, the internal architecture of metamaterials with predefined macroscopic shapes (e.g. 5x5x5 cube) can be designed for a wide range of mechanical deformations, from motions with unique degrees of freedom to motions with multiple (up to 3) degrees of freedom.

Minor clarification: Our approach can design for any combination of DOFs from none up to six since six is the maximum number of independent twists that any 3D system can possess. Hence the reason for the 7 columns of the FACT library. Note that we provide a 4 DOF example (Fig. 7i) and a 6 DOF example (Supplementary Fig. 4i).

Although their design strategy is not new (it was already developed for compliant mechanisms), its generalization in the context of metamaterials is novel and a very significant addition to the metamaterial toolbox. I believe that the results reported here will be of strong interest to mechanical engineers, physicists and material scientists interested in metamaterials. The off-the-self computational tool they provide might even make it readily useable beyond the scientific community, e.g. by product designers. Therefore, I think that this manuscript is a potential very good fit for Nature Communications. That being said, I think that as it stands, the paper lacks too much precision and clarity to be fully convincing and to be readable by the broad readership of Nature Communications. For these reasons, I strongly encourage the authors to address the following comments:

1. The extensive use of acronyms (more than 1 per line on average) makes the paper extremely difficult to follow. I believe that a significant rephrasing would allow the authors to use less repetitions and therefore to no rely that much on acronyms.

This is a great point. We eliminated the three most confusing and frequently used acronyms throughout the manuscript (i.e., freedom space (FS), intermediate freedom space (IFS), and constraint space (CS)) and more clearly phrased sentences as suggested. These changes were too numerous to highlight but you’ll see much fewer acronyms and clearer sentences where these words are used.

2. In some instances, the vocabulary is not precise enough, e.g. in the definition of “the yellow shaded region...” (l 143, p7), what is a “viable topology”? Also, always referring to the labelled yellow region is extremely confusing because as a reader, we lose track of the underlying scientific nature of such class of freedom spaces.

We clarified what we mean by “viable topology” in the text (pp. 7), and gave the spaces within the yellow region a technical name (i.e., cell spaces) to clarify their scientific significance. All instances referring to the yellow shaded region were altered accordingly (pp. 7, 8, 12, 14, 15, 16, 18, 20, 24, and 30).

3. In all the FEA plots (Fig. 3b, Figs.4g,h,i, figs. 5d,e,f, Figs. 6i,j, Fig. 7i, Figs. 8 b,c,e,f,g), it is never clearly stated in the caption which numerical protocol has been used to obtain these results. I understand from the text description of Fig. 4 that modal analysis has been used, but what about for the other figures? In general, the description of the FEA is too scarce to assess their validity and to reproduce them, e.g. the description of boundary conditions, mesh size, constitutive models are absent.

Another very important point! Thanks for catching this. We added details about all the FEA in the paper in two new Methods sections (pp. 10, 11, 18-19, 30 and 34-35, 38). We also added a clause in the data availability section about the CAD files necessary to replicate the results (pp. 35).

4. The description of the metamaterial geometric feature is incomplete: how are the beams and blades thickness chosen? How does this choice determine the compliance for target compliant deformations and for target non-compliant deformations.

The power of our approach is that it generates topologies (i.e., the kind, number, location, and orientation of flexible elements within a DCM) without the complexities of geometry considerations (e.g., lengths, widths, and thicknesses of the elements) or constituent material properties. This simplification is the secret sauce that enables rapid design generation. Topology is such a dominant factor in determining directions of compliance that regardless of constituent material properties or the geometry specified (as long as wires still look like wires (i.e., they are much longer than their diameter) and blades still look like blades (i.e., they are much thinner than they are long or wide)) the resulting DOFs remain largely unaffected. However, that being said, for our automated tool to draw our designs and conduct the modal analysis, the tool does need to be assigned material properties and the diameter of the wires. You'll notice these are asked for in the GUI during the design process. The reason that it's not discussed in the paper though, is that the topology (i.e., how many wires and their locations and orientations within the serially stacked layers) is calculated without material properties or wire diameters being specified. This is at the heart of why our computational approach is so disruptively fast at generating designs.

The consequence of this rapid design capability has trade offs though. We can't, for instance, initially set actual target stiffness values in the various directions. We can only guarantee that we will generate a topology that will polarize the compliant directions specified from the constrained unwanted directions so that there is as large a difference in stiffness as possible between them once material properties and geometries are assigned to the topology. Once a polarized topology has been generated though, an optimizer could be applied to determine the optimal diameter and material property of each wire to achieve target values of stiffness and compliance. But that's a task for a different and much less impactful optimization paper.

See pp. 6-7 for a discussion of this in the paper.

5. In general I feel that despite the generic design approach for the nature (wire, blade, etc..) and orientation of the slender elements, many design choices remain arbitrary and poorly explained. For instance, In fig. 2.: what sets the choice of five wires per unit cell? In fig. 4.: what sets the

choice of two blades per unit cell? in fig 4, from cell to cell, the blades are aligned in the vertical direction but not in the horizontal direction: why such a difference?

The answers to your questions are addressed in the Methods section, which are referenced in the main text for each example. The number of wires or blades that should be used to synthesize each cell such that resulting cells achieve their intended DOFs only and are as close to exactly constrained as possible is discussed on pp. 12, 27-30. The purpose of Supplementary Fig. 7 is to provide instructions for selecting elements from constraint spaces and the purpose of Supplementary Fig. 8 is to give examples of how to choose the kind (e.g., wires, blades, circular hyperboloid elements, etc.), number, location, and orientation of elements within constraint spaces to generate correct cells. See also pp. 25-26.

As for the vertical vs, horizontal question, both designs will achieve the same DOF if their elements are selected from the correct constraint spaces using the rules described in Methods, which they both are. Thus, from a FACT stand point, they achieve the same DOF which is infinitely compliant compared to the infinitely stiff constrained directions. In practice they of course differ with actual finite stiffness values, but generating topologies rapidly that are in the general stiffness and compliant ball park without these considerations is a huge benefit.

6. To convincingly demonstrate that the metamaterials are directionally compliant, it would be appropriate to provide measurements of the non-compliant deformation modes, e.g. in figure 9.

Although in principle I believe this would have been a good idea, there are an infinite number of other directions both translational and rotational in nature, which were designed to be constrained. The question is which of these should we have measured to compare against the compliant directions and why would we choose any of those over any others to compare? There's no good answer to these questions and we can't measure all the infinite options. This is why we chose to use modal analysis to verify directions of compliance throughout the majority of the paper's examples because the lowest natural frequency mode shapes are associated with the most compliant directions (i.e., DOFs). By providing lowest frequency mode shapes we demonstrate that all other infinite directions are effectively stiffer than those, whatever those stiffness values may be.

If we were to measure the stiffness in other directions on the DCM of Fig. 9 we would have needed to take those measurements when the other directions were measured because the material that is 3D printed changes over time as it is exposed to natural light. So, we'd have to re-print and retest the entire design, which would be very costly, time-consuming, and we believe would be unnecessary given all the other modal analysis verification in the paper.

7. About the counting of constraints and degrees of freedom in formula (4) of the methods: it is well known that the effects of multiple constraints can be redundant if they are related by symmetries, for instance when multiple bars are aligned (See Calladine, IJSS 1978). How do the authors treat such issue?

We only shy away from over-constraint when we are synthesizing each cell to reduce computational time and effort because exactly constrained cells require the placement of less elements than over-constrained cells by definition. We do not shy away from over-constraint in general though because every layer of cells is intended to massively over-constrain the final

DCM. This is why DCMs can be shaped in any way desired to still achieve the target DOFs because each unit cell redundantly constrains the system to achieve those DOFs. Thus, we leverage over-constraint to enable shape versatility. A detailed discussion about how to deal with exact and over constraint is provided on pp. 27-30.

8. So far, the design approach has exclusively focused on kinematics. However, I believe that the approach of the authors could easily be extended using e.g. beam and plate theory to also determine/design for a target compliance. This is probably outside of the scope of this work, but I am curious to hear the authors' take on this and I think that this aspect would be worth mentioning.

This paper focuses on kinematics because kinematics requires much less computation to deal with than kinematics combined with elastomechanics like most other approaches must deal with. But yes, your idea is fantastic and would provide a worthwhile extension to our theory for a future paper. Although we haven't incorporated plate theory (the part that excites me), we have in a way begun to incorporate beam theory when we construct our stiffness matrix in our custom modal analysis approach to verify our DOFs as discussed in the new section we added (pp. 34-35). Note that we can currently use our stiffness matrix to calculate specific stiffness values in any desired direction.

9. A related question concerning fig 9: I understand that the authors designed for the X and Y direction to be compliant, but they report very similar values of compliances in the X and Y directions. I find this result striking. Did the authors design for matching the X and Y compliances or is this just a coincidence? If it has been designed, the authors should explain how.

No, we didn't design for it specifically and we were also surprised to see how similar they ended up being. After some thought though we convinced ourselves that the reason why is that the relative dimensions between the parent design and the child cell design are the same (i.e., if a single scale factor were multiplied to all the dimensions in the small child cell, it would look identical to the homogenous parent design shown in Fig. 9a. Thus, since a smaller version of the design was rotated 90 degrees inside of a larger version of itself, I suppose we shouldn't be too surprised that they have the same stiffness although this interests me a lot. It may be a scientific principle that we write a future paper about regarding hierarchical metamaterials if we can prove it for general scenarios. We didn't want to discuss it in this paper, since it's not the focus of the paper, the paper is already very lengthy, and we aren't yet comfortable making any definitive statements about the underlying principle or reason for the similarity.

10. Could the authors comment of the importance of the nonlinear response? Since their structures comprise many slender elements, I would expect strong geometric nonlinearities to play a role.

Nonlinearities should only be considered once a DCM actually deforms a finite amount. Directions of compliance exist without anything ever needing to deform at all (just like a spring can be considered stiff whether or not it is loaded). Since we are designing DCMs that achieve desired directions of compliance before anything is ever deformed, we don't need to consider

finite deformation nonlinearities, especially during the design process. Such nonlinearities would substantially complicate the underlying mathematics and would again make it impossibly slow and impractical to manage the design of such materials. The advance of this paper came by stripping out all unnecessary considerations and focusing on topology and linear mathematics only.

11. What determines the optimal choice for the number of unit cells?

It depends on the desired shape of the final DCM. The more cells in the desired volume, the more bulk shapes you'll be able to access that simultaneously achieve the desired DOFs, but the more computational time and effort is necessary. A discussion of this is provided on pp. 9, 12, 14, and 23.

Thank you for all the wonderful suggestions. We feel that the paper is now much stronger and we hope we've cleared up your concerns.

Reviewer #2:

A new computational tool was created to design directionally compliant metamaterials more efficiently, since design of metamaterials is complex due to the number and locations of flexible elements. The new software tool takes orders of magnitude less time (tens of seconds vs tens of hours) than current computational tools for this application.

This computational model was based on the Freedom and Constraint Topologies (FACT) methodology with some simplified assumptions in order to more efficiently compute designs. The simplified assumptions are that only geometry and orientation are considered in designing and the beams are infinitely stiff along the constraint-force lines but compliant in all other directions.

Basic rules for use of the FACT methodology and mathematical definitions to model the methodology within the tool were presented to explain how the tool determines the design. Case studies were performed on single degree-of-freedom (DOF), multi-DOF, multi-DOF with CS outside the yellow shaded region, and bulk shape systems to show the design works for many different cases.

The discussion on the FACT methodology used in each case studies along with the accompanying figures clearly shows the process by which the computational tool efficiently designs these compliant metamaterials. This helped me to better understand how the methodology was applied to design the computational tool.

There is an impressive variety of case studies presented. This showed that the tool has been tested and can be used for many different designs. The supplementary information added even more case studies to show that this tool works for many different cases.

CHANGES THAT SHOULD BE MADE BEFORE PUBLICATION

At Line 435, "existing" should be "exiting".

Thank you for catching this. We made the change (pp. 25).

CHANGES THAT SHOULD BE CONSIDERED TO IMPROVE THE PAPER

The flow and readability of Figure 2 could be improved by placing 2d, 2e prior to 2c. This would give the figure a more linear feel as to how the process works.

We see how this could be confusing if each part of the figure were intended to represent each step of the process. Since this is the first example of the paper, however, we wanted to provide a high-level picture of the approach, which is actually represented by parts a through c only (a is the empty template volume that will be filled with elements (input), b is the approach that fills the space with elements, and c is the final result (output)). Parts d and e are simply describing the parts of the final design and we feel that it's important to show the final design first before we begin to describe what it's made of. We decided to add arrows between a and b and b and c to clarify this concept in Fig. 2.

Suggest placing the comments about scale bars and color definitions in the corresponding subfigure description in Figures 3, 6, 7, and 9. For example, for Figure 3 – Place “(scale bar in a, 50 μm)” at the end of the description of a. As another example, for Figure 6 – Place “colors in i and j are defined in Fig. 3b” at the end of the description of i, j.

We put the scale bar and color definition at the end of the figure captions to conform with *Nature Communications* formatting guidelines. At least all other paper examples we saw that are published in *Nature Communications* puts these at the end of the figure caption.

Lines 165 – 170 discuss that layer extensions may be added to the design. Is there any time where these extensions could interact with a different layer and would thus affect the desired compliance of the design? In looking at this in the context of the rest of the paper I don't think there would be interactions but discussing that the layer extensions won't affect the desired compliance may be helpful.

This is a very insightful comment and an interesting question. The extensions would definitely have an effect on the stiffness of various DCMs, but they aren't a dominant factor in polarizing directions of compliance and stiffness. The topology alone dominates in determining which directions will be compliant and which will be stiff. All other considerations (e.g., geometry of elements, constituent material properties, and geometry of stage extensions) would determine the specific stiffness values and how different they are from each other. We leverage this observation and only consider the topology when designing DCMs so we can quickly generate topologies that are guaranteed to have the DOFs be the most compliant directions. Then all other considerations could be optimized to tune their actual values. Since the paper is already very lengthy, we don't feel that a discussion of this rises to the level of a necessary discussion in the paper.

Thank you for taking the time to review this paper to help make it better.

REVIEWERS' COMMENTS:

Reviewer #1 (Remarks to the Author):

Dear Editor,

the authors have successfully addressed most of my comments. The paper is now clearer and more precise.

A minor point: I want to follow up on my question 4: I understand that the authors do not want to mention beam/blade geometry for the design approach. However, I could not find the values used by the authors for their real 3D printed samples. I believe that mentioning these values could be very useful to the interested reader: 3D printing such large aspect-ratio structural details is likely to be the main fabrication bottleneck.

Reviewer #2 (Remarks to the Author):

The authors have done a nice job responding to the reviews. Here is one minor note to consider before publication:

Pg 31 Line 583, says "perform the linear modal analyses for the case studies of Figs. 4g-i, 5d-e", but looks to me like it should be "5d-f"

Response to Reviewers

Reviewer #1 (Remarks to the Author):

Dear Editor,

the authors have successfully addressed most of my comments. The paper is now clearer and more precise.

A minor point: I want to follow up on my question 4: I understand that the authors do not want to mention beam/blade geometry for the design approach. However, I could not find the values used by the authors for their real 3D printed samples. I believe that mentioning these values could be very useful to the interested reader: 3D printing such large aspect-ratio structural details is likely to be the main fabrication bottleneck.

Thank you for your helpful feedback in making the paper clearer and more precise. As to your point, we set the diameters of the wire elements and the thicknesses of the blade elements in all of our real 3D printed samples to be the smallest feature size our printer was comfortable printing. The blade widths were made as wide as possible within the space they had available within their respective cells so they would behave as much like an ideal blade as possible. All element lengths were determined by the distance between the DCM layers and their angles. We decided that because there were so many geometric parameters that defined each cell within each 3D printed design that instead of significantly lengthening the paper by providing all of these parameters for every design that we printed (and possibly derailing the focus of the paper since FACT operates independent of these parameters), we would provide CAD or .stl files of the printed structures upon the reader's request and declare that in our data availability statement. Of course all the actual geometric parameters used are embedded in those files if any reader did want to exam them. Such information would be more important to provide for case studies directly within a follow-on paper that optimizes the geometry of the topologies that FACT generates.

Reviewer #2 (Remarks to the Author):

The authors have done a nice job responding to the reviews. Here is one minor note to consider before publication:

Pg 31 Line 583, says "perform the linear modal analyses for the case studies of Figs. 4g-i, 5d-e", but looks to me like it should be "5d-f"

Good catch! We made the change in track changes. Thank you again for your help.